# Aging power spectrum of membrane protein transport and other subordinated random walks

Zachary R. Fox [1,2], Eli Barkai[3] & Diego Krapf [1,4✉]

Single-particle tracking offers detailed information about the motion of molecules in complex environments such as those encountered in live cells, but the interpretation of experimental data is challenging. One of the most powerful tools in the characterization of random processes is the power spectral density. However, because anomalous diffusion processes in complex systems are usually not stationary, the traditional Wiener-Khinchin theorem for the analysis of power spectral densities is invalid. Here, we employ a recently developed tool named aging Wiener-Khinchin theorem to derive the power spectral density of fractional Brownian motion coexisting with a scale-free continuous time random walk, the two most typical anomalous diffusion processes. Using this analysis, we characterize the motion of voltage-gated sodium channels on the surface of hippocampal neurons. Our results show aging where the power spectral density can either increase or decrease with observation time depending on the specific parameters of both underlying processes.

[1] School of Biomedical Engineering, Colorado State University, Fort Collins, CO, USA. [2] The Center for Nonlinear Studies and Computational and Statistical Sciences Division, Los Alamos National Laboratory, Los Alamos, NM, USA. [3] Department of Physics, Institute of Nanotechnology and Advanced Materials, Bar-Ilan University, Ramat-Gan, Israel. [4] Electrical and Computer Engineering, Colorado State University, Fort Collins, CO, USA. ✉email: diego.krapf@colostate.edu

A very large class of biological and physical systems exhibit correlations that extend across multiple time scales. This feature is also found in social networks as well as in complex systems made of interacting components like glasses. Such correlations manifest themselves as a broad spectrum of relaxation times and in the practically universal emergence of $1/f$ decay in the power spectrum, which points to self-similarity in the dynamics at different timescales[1,2]. The effect is predominantly found at low frequencies where the contributions of each frequency $\omega = 2\pi f$ to the overall power spectral density (PSD) exhibit a power law $S(\omega) \sim 1/\omega^\beta$, with $0 < \beta \le 2$[3–7]. To name a few diverse examples, $1/f$ spectra are observed in nanoscale devices[8,9], network traffic[10], earthquakes[11], heartbeat dynamics[12], DNA base sequences[13], climate[14], and ecology[15]. Mandelbrot and later Bouchaud et al. suggested that the processes involved are inherently non-stationary leading to the idea that the spectrum should depend both on the frequency and the measurement time[5,16,17]. Indeed, the very basic formula describing these ubiquitous phenomena was recently replaced with a more general one[6]. Based on experimental data of blinking quantum dots[18], nanoelectronic devices[9,19], and fluctuations of interfaces[7], the basic spectrum must be described with a new formula $S(\omega, t_m) \sim \omega^{-\beta} t_m^z$, where $t_m$ is the measurement time. These developments, in turn, motivated a new theoretical framework, called aging Wiener–Khinchin theorem[20–22]. This new theorem replaces the celebrated Wiener–Khinchin theorem valid for stationary processes, which is widely applicable to systems that do not exhibit $1/f$ noise[23].

Notwithstanding previous advances, many questions remain open. First, the aging Wiener–Khinchin theorem relates the aging power spectrum with $z \ne 0$ to a non-stationary correlation function (soon to be discussed). However, how can one find this correlation function? As for the standard Wiener–Khinchin theorem, the correlation function is specific to the system. In the context of diffusion in cells as well as in many other complex systems, Mandelbrot's fractional Brownian motion (fBM)[24] and the Montroll–Weiss continuous time random walk (CTRW)[25] are two widely investigated models of anomalous transport. While the fluctuations in fBM are stationary, the CTRW process is inherently non-stationary. However, both models, when standing alone, are usually non-sufficient to describe the transport of particles that alternate between a trapping phase (like in CTRW) and correlated motion (like in fBM), as is the case in live cells, for example due to interactions in a viscoelastic medium[26]. The open questions begin with how to create a marriage between these models? Then, can we obtain the correlation functions and $1/f$ spectrum? Achieving these goals will show how the exponents $\beta$ and $z$ depend on the underlying processes, and will determine which of the processes is dominating the PSD. Finally, most importantly, these goals can elucidate whether the whole approach to the PSD is useful in experiments. Specifically, we demonstrate the applicability of aging Wiener–Khinchin theorem and the corresponding calculation of the correlation function with experimental recording of the power spectra of the motion of ion channels in the plasma membrane of mammalian cells.

The emergence of $1/f$ noise has triggered notable interest in biological environments both from a fundamental point of view and due to its relevance in pathologies and disease[27]. Self-similar temporal characteristics are observed in biological systems of broadly different length scales. Recent molecular dynamics simulations in combination with previous experimental results have shown that the internal dynamics in globular proteins are self-similar and the autocorrelation function is aging over an astonishing 13 decades in time[2,28]. These fluctuations play essential roles in cell functions that involve molecular interactions such as gene regulation. In fact, this behavior is widespread and found from the dynamics of proteins within cell membranes to the scaling behavior of heartbeat time series[27,29]. Nevertheless, it still remains a challenge to measure how aging affects the spectrum of recorded $1/f$ noise in real systems.

Single molecule tracking in the cell environment has been used extensively to shed light on the functions and interactions of the molecules that make life possible[30–33]. Spectral analyses are emerging as a key tool in the characterization of individual molecule trajectories in biological systems because it informs on features that are difficult to infer using other traditional statistics[6,34–38]. It has been observed that among traditional statistical approaches, e.g., analyses based on the mean squared displacement, the PSD appears to be less sensitive to external noises[39]. Following previous work, we promote a theory that shows how the most basic formula of $1/f$ noise needs modifications, namely that $S(\omega, t_m) \sim \omega^{-\beta} t_m^z$ as mentioned. The question that still needs to be addressed is what the physical meaning of the new exponent $z$ is, to explore cases where it is negative (corresponding to a decrease of the PSD with time and, hence, aging) and cases where it is positive (corresponding to a PSD increasing with time and, hence, rejuvenation). Further, beyond the development of the theory, it is important to show how these effects are found experimentally.

Traditionally, the PSD of a time-dependent signal is defined as the average over an infinitely large ensemble in the limit of infinite time (Supplementary Eq. 1). In practice, when analyzing either experiments or numerical simulations, one does not have access to infinite measurement time, nor to a large ensemble of trajectories, and the PSD is estimated by using the periodogram. For stationary processes, the PSD can be directly calculated from the autocorrelation function, using the relation provided by the Wiener–Khinchin theorem (Supplementary Eq. 2)[23]. The Wiener–Khinchin theorem holds for a large class of time-invariant processes, where the concept of a time-independent limiting power spectrum is useful. One could wonder how to extend the Wiener–Khinchin theorem to non-stationary processes, but, due to the extensive variety of such processes, this general approach appears a priori to be a futile direction of research. Nonetheless, this first assessment turns out to be wrong. There exists a large class of stochastic processes describing systems that are non-stationary but scale invariant. Specifically, the autocorrelation function explicitly depends on time $t$ via the expression $C_{EA}(\tau, t) = \langle x(t)\, x(t + \tau) \rangle \sim t^\gamma \phi_{EA}(\tau/t)$, where $\phi_{EA}(\tau/t)$ is a scaling function. As mentioned, a new theoretical framework was developed for this very large class of scale invariant processes, the aging Wiener–Khinchin theorem[20–22]. The PSD that emerges in this case is, in turn, directly related to $1/f$ noise and depends on the observation time.

Here, we address the spectral content of processes with scale free relaxation times, using both theoretical modeling and experimental validation. We show how the aging Wiener–Khinchin theorem is a useful tool and, more importantly, demonstrate how the aging exponent $z$ and the spectral exponent $\beta$ are related to the underlying processes. To reach this goal, we obtain the non-stationary correlation function of the subordinated fBM, which combines two well known approaches to anomalous diffusion. Depending on whether the process is negatively or positively correlated, we get vastly different frequency decays of the power spectrum. Thus, the aging Wiener–Khinchin theorem can be used to classify widely different classes of dynamics. Finally, by analyzing the dynamics of voltage-gated sodium channels (Nav) on the somatic membrane of hippocampal neurons, we demonstrate the usefulness of the approach, and prove that its basic principles work in the laboratory. These experimental data reveal how one can use a few long trajectories and estimate the exponents characterizing the dynamics with high precision. Our work, thus, not only validates the aging Wiener-Khinchin

theorem as an emerging tool in spectral analysis, but it also unravels the meaning of the exponents describing the aging and the frequency decay.

## Results

**Aging Wiener–Khinchin theorem.** In any stationary process $x(t)$, the PSD is related to the autocorrelation function (ACF) $C_{\mathrm{EA}}(\tau) = \langle x(t)x(t+\tau)\rangle$ via the fundamental Wiener–Khinchin theorem (Supplementary Eq. 2). Throughout the manuscript we employ the subscripts EA and TA to denote ensemble averages and time averages, respectively. However, diffusive processes are intrinsically non-stationary and thus the Wiener–Khinchin theorem is invalid. In recent years, power spectrum theory has been expanded with a tool called the aging Wiener–Khinchin theorem[20–22]. This theorem covers a broad class of non-stationary processes that possess an auto-correlation function with the long-time asymptotic $C_{\mathrm{EA}}(t, \tau) = \langle x(t)x(t+\tau)\rangle \sim t^\gamma \phi_{\mathrm{EA}}(\tau/t)$. Such correlation functions are common[22,40,41] and they are called scale invariant. An alternative analysis of the autocorrelation function is performed in terms of its time average $C_{\mathrm{TA}}$ of individual trajectories, where

$$C_{\mathrm{TA}}(t_{\mathrm{m}}, \tau) = \frac{1}{t_{\mathrm{m}} - \tau} \int_0^{t_{\mathrm{m}}-\tau} x(t)x(t+\tau)\mathrm{d}t, \qquad (1)$$

with $t_{\mathrm{m}}$ being the measurement time. For ergodic processes, $C_{\mathrm{TA}}$ converges to $C_{\mathrm{EA}}$ in the long time limit. However, when the process is not ergodic, such as a scale-free CTRW, $C_{\mathrm{TA}}$ of individual trajectories remain random variables even in the long time limit[42,43]. Thus, one analyzes the ensemble-average of the TA-ACF, $\langle C_{\mathrm{TA}}(t_{\mathrm{m}}, \tau)\rangle$. Further, ergodicity breaking leads to a difference in the two averages, $\langle C_{\mathrm{TA}}(t_{\mathrm{m}} = t, \tau)\rangle \neq C_{\mathrm{EA}}(t, \tau)$. Each of these formalisms (ensemble vs. time averages) has its own advantages and disadvantages. Nevertheless, when the number of trajectories is small and the measurement time is long, the time averages lead to better statistics and it is, thus, the more commonly used method in single-particle tracking. When $C_{\mathrm{EA}}(t, \tau) = t^\gamma \phi_{\mathrm{EA}}(\tau/t)$, the time-average ACF has also the scaling form $\langle C_{\mathrm{TA}}(t_{\mathrm{m}}, \tau)\rangle = t_{\mathrm{m}}^\gamma \phi_{\mathrm{TA}}(\tau/t_{\mathrm{m}})$[20]. The scaling function $\phi_{\mathrm{TA}}(\tau/t_{\mathrm{m}})$ is directly related to the ensemble average via the relation

$$\phi_{\mathrm{TA}}(y) = \frac{y^{1+\gamma}}{1-y} \int_{\frac{y}{1-y}}^{\infty} \frac{\phi_{\mathrm{EA}}(z)}{z^{2+\gamma}} \mathrm{d}z, \qquad (2)$$

where $y = \tau/t_{\mathrm{m}}$, which implies $0 \leq y \leq 1$.

For a measurement time $t_{\mathrm{m}}$ the power spectrum can be only obtained for the discrete set of frequencies $\omega_k t_{\mathrm{m}} = 2\pi k$ with $k$ being a non-negative integer. That is, the frequencies can be resolved down to $\Delta\omega = 2\pi/t_{\mathrm{m}}$, which decays to zero in the limit of large measurement time $t_{\mathrm{m}}$. The aging Wiener–Khinchin theorem relates the average power spectrum for this set of frequencies to the time-averaged autocorrelation function[20,22],

$$\langle S(\omega, t_{\mathrm{m}})\rangle = 2t_{\mathrm{m}}^{1+\gamma} \int_0^1 (1-y)\phi_{\mathrm{TA}}(y)\cos(\omega t_{\mathrm{m}}y)\mathrm{d}y. \qquad (3)$$

A relation between the PSD and the ensemble-averaged correlation function also exists, but we will employ the relation to the time average because of its more common use in single-particle tracking experiments.

## The model for subordinated random walks. 

A useful way to model the diffusive transport in live cells is via the combination of two stochastic processes: the CTRW and fBM. On one hand, the CTRW constitutes the quintessential diffusion process with heavy-tailed immobilization times and has been extensively used to describe transport in disordered environments[44,45], protein dynamics in mammalian cells[30–32,46], and even to model financial markets[47]. On the other hand processes with correlated increments such as fBM or diffusion in fractal environments have been

often observed to lead to anomalous transport with memory effects[48–50]. fBM is the only Gaussian self-similar process with stationary increments, of which Brownian motion constitutes a special case. Technically the combination of these widely observed models is made possible with a subordination technique[32,51,52]. In a subordination scheme, the steps of a random walk take place at operational times $t_n$ defined by a directing stochastic process. For example, antipersistent motions accompanied by heavy-tailed immobilization times, have been observed in live cells in the motion of ion channels[53], insulin granules[54], membrane receptors[55], and nanosized objects in the cytoplasm[56], as well as for tracer particles in actin networks in vitro[57]. Sub-ordinated processes constitute one of the most general classes of random walks and are widespread beyond the dynamics in the cell[4,7,19,58]. This scheme allows to evaluate processes with short-range or long-range memory and non-stationarity, leading to complex aging properties.

We consider a fBM-like process at discrete times, $n = 0, 1, 2, 3,\ldots$, with Hurst exponent $H$, such that its autocorrelation function at the discrete times $n$ is given by[24]

$$\langle x_n x_{n+\Delta n}\rangle = \Delta x^2 \big[n^{2H} + (n+\Delta n)^{2H} - \Delta n^{2H}\big], \qquad (4)$$

where the coefficient $\Delta x$ is a scaling parameter with units of length. We place the process defined by Eq. (4) under the operational time of a CTRW, so that the particle is immobilized during sojourn times with a heavy-tailed distribution. Such immobilizations arise, for example, from energetic disorder where a particle has random waiting times at each trapping site[25,29,59,60].

The operational times are defined by a random process $\{t_n\}$ with non-negative independent increments $\tau_n = t_n - t_{n-1}$. The time increments $\tau_n$ between renewals are, in the long time limit, asymptotically distributed according to a probability density function[61]

$$\psi(\tau_n) \sim \frac{\alpha}{\Gamma(1-\alpha)} \frac{t_0^\alpha}{\tau_n^{1+\alpha}}, \qquad (5)$$

where $0 < \alpha < 1$, $t_0$ is a constant with units of time, and $\Gamma(\cdot)$ is the gamma function. At time $t$, the position of the particle is $x(t) = x_n$ where $n$ is the random number of renewals in the interval $(0, t)$. Given $n$, the position $x_n$ is determined by the discrete fBM process defined by Eq. 4. Three representative trajectories of such a process are shown in Fig. 1. The ensemble-averaged autocorrelation function of $x(t)$ is then

$$\begin{aligned} C_{\mathrm{EA}}(t, \tau) &= \langle x(t)x(t+\tau)\rangle \\ &= \mathbb{E}\big[\mathbb{E}[x(t)x(t+\tau)|n_t; (n+\Delta n)_{t+\tau}]\big], \end{aligned} \qquad (6)$$

where $\mathbb{E}[g(x)] = \langle g(x)\rangle$ represents the expected value of $g(x)$ and $\mathbb{E}[g(x)|y]$ is the conditional expected value of $g(x)$ given $y$. In particular, the last term indicates the iterated expectation of $x(t)x(t+\tau)$, given that $n$ steps have taken place up to time $t$ and $n + \Delta n$ steps have taken place up to time $t + \tau$. Further, we define $\chi_{n,\Delta n}(t, \tau)$ as the joint probability of taking $n$ steps up to time $t$ and $\Delta n$ steps in the interval $(t, t + \tau)$. Combining Eqs. (6) and (4), we obtain

$$\begin{aligned} C_{\mathrm{EA}}(t, \tau) &= \mathbb{E}\Big[\Delta x^2\Big(n_t^{2H} + (n+\Delta n)_{t+\tau}^{2H} - \Delta n_{\tau,t}^{2H}\Big)\Big] \\ &= \Delta x^2 \sum_{n=0}^{\infty} \sum_{\Delta n=0}^{\infty} \big(n^{2H} + (n+\Delta n)^{2H} - \Delta n^{2H}\big)\chi_{n,\Delta n}(t, \tau). \end{aligned}$$
$$(7)$$

Once the ensemble-averaged autocorrelation function is found, we can obtain the time-averaged $C_{\mathrm{TA}}(t_{\mathrm{m}}, \tau)$ via Eq. (2) and, subsequently, the PSD using the aging Wiener–Khinchin theorem (Eq. 3).

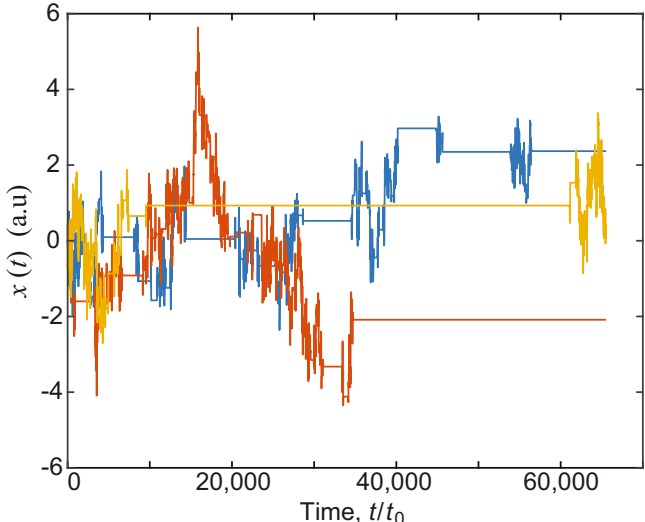

**Fig. 1 Representative trajectories for a subordination fractional Brownian motion process.** The Hurst exponent in these trajectories is $H = 0.3$ and the CTRW anomalous exponent is $\alpha = 0.8$. Long immobilization times are observed within the fractional Brownian motion.

**Continuous time random walk ($2H = 1$).** The fBM reverts to Brownian motion when $2H = 1$ and, thus, the process becomes a traditional CTRW[25,62]. The ensemble-averaged autocorrelation function in Eq. (7) becomes (see Supplementary Note 3)

$$C_{EA}(t, \tau) \sim \frac{2\Delta x^2}{t_0^\alpha \Gamma(1 + \alpha)} t^\alpha, \quad (8)$$

which, given the memoryless property of Brownian motion, boils down to the ensemble-averaged autocorrelation function being independent of lag time $\tau$ and equal to the mean squared displacement (MSD), $C_{EA}(t, \tau) = 2\Delta x^2 \langle n(t) \rangle = \langle x^2(t) \rangle$. The MSD solution for the CTRW is $\langle x^2(t) \rangle \sim t^\alpha$, that is, it exhibits sub-diffusion with anomalous exponent $\alpha$[61].

The ensemble-averaged autocorrelation function in Eq. (8), for $2H = 1$, implies that $C_{EA} = t^\alpha \phi_{EA}$ with $\phi_{EA}$ being a constant. The time-averaged autocorrelation function is $\langle C_{TA} \rangle = t_m^\alpha \phi_{TA}(\tau/t_m)$ and we find (Supplementary Eq. 17)

$$\langle C_{TA}(t_m, \tau) \rangle = \frac{2\Delta x^2 t_m^\alpha}{t_0^\alpha \Gamma(2 + \alpha)} \left( 1 - \frac{\tau}{t_m} \right)^\alpha. \quad (9)$$

Next, we use the time-averaged autocorrelation function (Eq. 9) in conjunction with the aging Wiener–Khinchin theorem to obtain the PSD of the CTRW. We find the exact solution of the sample power spectral density by solving the integral in Eq. (3). The PSD (Supplementary Eq. 18) is a function of both frequency $\omega$ and realization time $t_m$. Expanding the PSD for $\omega t_m \gg 1$, it is found that the leading term scales in frequency as $\omega^{-2}$ and in time as $t_m^{-(1-\alpha)}$,

$$\langle S_{2H=1}(\omega, t_m) \rangle \sim \frac{4\Delta x^2}{t_0^\alpha \Gamma(1 + \alpha)} \frac{1}{t_m^{1-\alpha} \omega^2}, \quad (10)$$

which is related to the MSD via the relation

$$\langle S_{2H=1}(\omega, t_m) \rangle \sim \frac{2}{\alpha \omega^2} \frac{\partial}{\partial t_m} \langle x^2(t_m) \rangle. \quad (11)$$

This is a useful relation that connects the fluctuations in the trajectory (the PSD) to transport properties (the MSD) for the CTRW. Importantly, the MSD is proportional to the mean number of renewals, thus Eq. (11) provides a connection between the PSD and the number of renewals. While Eq. (11) applies to

the CTRW, we will see later that it is not universal for the scale free processes under study.

Figure 2 shows a comparison of these analytical results to numerical simulations of 10,000 realizations with $\alpha = 0.7$. The MSD exhibits a power law, $\langle x^2(t) \rangle \sim t^\alpha$ (Fig. 2a). The power spectral density is presented in Fig. 2b for five different measurement times from $t_m = 2^8$ to $2^{16}$ and shows good agreement with the power law asymptotic $\omega^{-2}$. As shown in Supplementary Note 3, using hypergeometric functions we can get the exact PSD; however, the power law asymptotics show highly accurate results. The spectra also exhibit aging with an amplitude that scales as $t_m^{-(1-\alpha)}$ (Fig. 2c). Intuitively, as the measurement time increases, we encounter longer stagnation periods and, hence, the PSD decays with measurement time. Physically, this effect is due to the broadly distributed trapping times in the system.

**Subordinated process involving fBM ($0 < H < 1$).** We now deal with subordinated random walks where the increments exhibit correlations. When $2H \neq 1$, the process has positively correlated increments for $H > 0.5$ and negatively correlated increments for $H < 0.5$. The autocorrelation function $C_{EA}$ in Eq. (7) is

$$C_{EA}(t, \tau) = \Delta x^2 \left[ \langle n^{2H}(t) \rangle + \langle n^{2H}(t + \tau) \rangle - \langle \Delta n^{2H}(\tau; t) \rangle \right], \quad (12)$$

where $\Delta n(\tau; t)$ is the number of steps between the aged time $t$ and $t + \tau$. Using renewal theory and the power law waiting time distribution in Eq. (5), the terms in Eq. (12) can be expressed via hypergeometric functions (Supplementary Eqs. 21 and 22). The ensemble-averaged autocorrelation function (Supplementary Eq. 25) has the form $C_{EA}(t, \tau) = t^\gamma \phi_{EA}(\tau/t)$, which implies the time-averaged autocorrelation function is of the form $\langle C_{TA}(t_m, \tau) \rangle = t_m^\gamma \phi_{TA}(\tau/t_m)$[22]. Following Eq. (2), we find the scaling function $\phi_{TA}(\tau/t_m)$. The exact analytical results for the time-averaged ACF (Supplementary Eq. 27) were compared to numerical simulations. The simulations are observed to agree with analytical results for both $H < 0.5$ and $H > 0.5$ in Supplementary Fig. (1a, b), respectively.

The calculation of the PSD with the correlation function involves two steps. Our approach uses the scale invariant correlation function, which was tested versus numerical results, and the aging Wiener–Khinchin theorem, Eq. (3). The calculation essentially leads to PSDs that are expressed in terms of hypergeometric functions (Supplementary Eq. 33) and can be simplified. The idea is to use the large frequency limit to obtain approximate results of the aging $1/f$ noise type. These work well, as we show later in the figures. By expanding the PSD in the limit $\omega t_m \gg 1$ and noting that the spectrum is evaluated at frequencies $\omega t_m = 2\pi k$, we obtain the leading term, which depends on the specific values of $\alpha$ and $H$. In the case that the increments are anticorrelated, i.e., $H < 0.5$,

$$\langle S_{H < 1/2}(\omega, t_m) \rangle \approx 2c t_m^{-(1-\alpha)} \omega^{-2+\alpha-2\alpha H}, \quad (13)$$

where $c$ is a constant defined explicitly in Supplementary Eq. (38). An example of this antipersistent case is shown for numerical simulations with $\alpha = 0.4$ and $H = 0.3$ in Fig. 3a. The scaling of the PSD both in $t_m$ and $\omega$ agrees with Eq. (13).

When the increments of the random walk are positively correlated (i.e, $H > 1/2$), the leading term is

$$\langle S_{H > 1/2}(\omega, t_m) \rangle \approx 2D t_m^{2\alpha H - 1} \omega^{-2}, \quad (14)$$

with $D$ being a generalized diffusion coefficient (Supplementary Eq. 26). This PSD is related to the mean square displacement in a

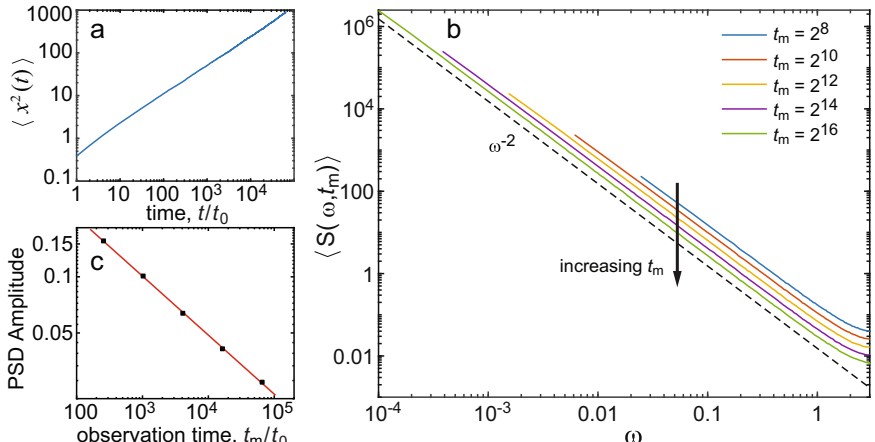

**Fig. 2 Numerical simulation of the CTRW, i.e., Brownian motion with power-law waiting times.** The simulations were performed for $\alpha = 0.7$ and 10,000 realizations were obtained. **a** The MSD shows subdiffusive behavior $\langle x^2(t) \rangle \sim t^\alpha$, while a linear regression of $\log(\mathrm{MSD})$ vs. $\log(t)$ indicates $\langle x^2(t) \rangle \sim t^{0.69}$. The times and displacements are unitless, i.e., the simulation sampling time is 1. **b** PSD at five different measurement times exhibits aging. The power law asymptotic $S \sim \omega^{-2}$ is indicated with a dashed line. The arrow shows the decay in the PSD with measurement time $t_\mathrm{m}$. **c** The amplitude $A(t_\mathrm{m})$ of the PSD, where $\langle S \rangle = A(t_\mathrm{m})/\omega^2$, shows $A(t_\mathrm{m}) \sim t_\mathrm{m}^{-0.31}$, highlighting the aging effect, in excellent agreement with theory which predicts $A(t_\mathrm{m}) \sim t_\mathrm{m}^{-(1-\alpha)}$.

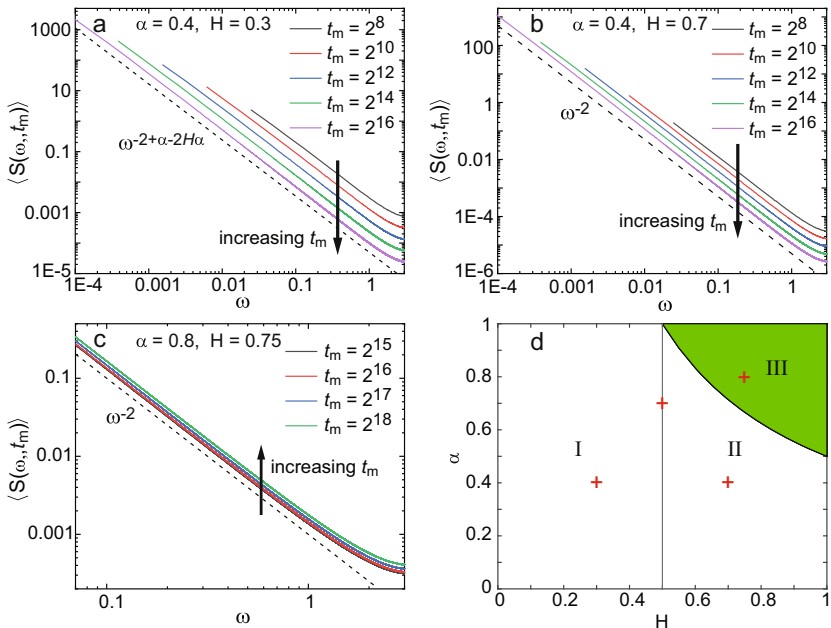

**Fig. 3 Power spectral density of numerical simulations of fBM with heavy-tailed immobilization times.** **a** Simulations for five different measurement times with $\alpha = 0.4$ and $H = 0.3$. The number of realizations is $N = 10,000$. Given that the fBM is subdiffusive ($H < 1/2$), the PSD is predicted to scale as $\langle S(\omega, t_\mathrm{m}) \rangle \sim t_\mathrm{m}^{-(1-\alpha)} \omega^{-2+\alpha-2\alpha H}$ as in Eq. (13). The dashed line shows the scaling $\omega^{-2+\alpha-2\alpha H}$ and the arrow indicates the decay in the PSD as the measurement time $t_\mathrm{m}$ increases. **b** Simulations for five different measurement times with $\alpha = 0.4$ and $H = 0.7$, $N = 10,000$ realizations. The fBM is superdiffusive ($H > 1/2$) and the PSD is, thus, predicted to scale as $\langle S(\omega, t_\mathrm{m}) \rangle \sim t_\mathrm{m}^{-(1-2\alpha H)} \omega^{-2}$ (Eq. (14)). The dashed line shows the scaling $\omega^{-2}$ and the arrow shows the decay in the PSD with measurement time $t_\mathrm{m}$. **c** Simulations for five different measurement times with $\alpha = 0.8$ and $H = 0.75$, $N = 5,000$ realizations. Given that $2\alpha H > 1$, the power spectrum increases with measurement time as indicated by the arrow. The dashed black line indicates $\omega^{-2}$. **d** The shaded region (regime III) indicates the set of values for $\alpha$ and $H$ that yields a PSD $\langle S(\omega, t_\mathrm{m}) \rangle$ that increases with measurement time. In the rest of the plane, the power spectrum decays with $t_\mathrm{m}$. Within this part of the plane, regime I is characterized by $\langle S(\omega, t_\mathrm{m}) \rangle \sim t_\mathrm{m}^{-(1-\alpha)} \omega^{-2+\alpha-2\alpha H}$ and regime II by $\langle S(\omega, t_\mathrm{m}) \rangle \sim t_\mathrm{m}^{-(1-2\alpha H)} \omega^{-2}$. The red crosses indicate the pairs $(H, \alpha)$ used in the examples in **a**–**c**, and the CTRW in Fig. 2.

similar way as the CTRW, via the relation

$$\langle S_{H > 1/2}(\omega, t_\mathrm{m}) \rangle \approx \frac{1}{2\alpha H \omega^2} \frac{\partial}{\partial t_\mathrm{m}} \langle x^2(t_\mathrm{m}) \rangle, \qquad (15)$$

which is similar to Eq. (11), albeit with a factor $1/2$. When $H > 1/2$, the PSD decreases with observation time for small $\alpha$ and $H$, namely when $2\alpha H < 1$. Otherwise (shaded regime III in Fig. 3d),

the PSD increases with observation time. Figure 3b shows the power spectra for numerical simulations where the underlying fBM is superdiffusive with $H = 0.7$ and $\alpha = 0.4$, which falls in the regime that $\langle S(\omega, t_\mathrm{m}) \rangle$ decays with $t_\mathrm{m}$ (regime II in Fig. 3d). Figure 3c shows simulations with $H = 0.75$ and $\alpha = 0.8$ where $\langle S(\omega, t_\mathrm{m}) \rangle$ indeed is observed to increase with $t_\mathrm{m}$. In this regime of increasing $S$, the convergence to Eq. (14) is very slow and appears

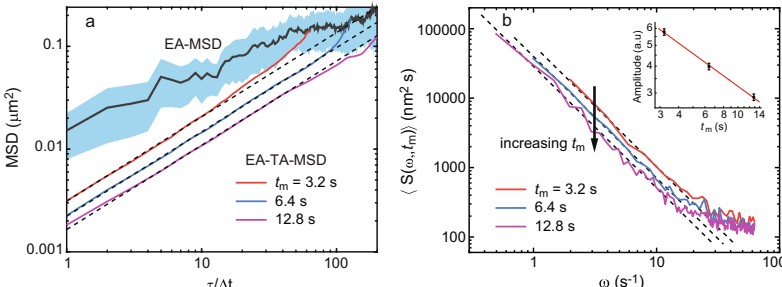

**Fig. 4 Analysis of Nav1.6 experimental trajectories in the soma of hippocampal neurons. a** The time-averaged MSD is different from the ensemble-averaged MSD (gray upper line). The shaded region indicates the 95% confidence interval for the ensemble-averaged MSD. The time-averaged MSD scales with the lag time as $\tau^{0.81\pm0.05}$ (dashed lines), while exhibiting aging as it decays with experimental time as $1/t_m^{1-\alpha}$, from which $\alpha$ is estimated to be $0.54 \pm 0.02$. **b** Average spectra are presented for three measurement times. The dashed lines show a scaling $1/\omega^{1.75}$. Besides the power-law scaling, the spectra exhibit white noise evident at large frequencies, likely due to localization error. The arrow shows the decay in the PSD with measurement time $t_m$. The inset shows the amplitude of the PSD as a function of measurement time in a log-log plot. It shows that the spectrum exhibits aging with a power law scaling $1/t_m^{1-\alpha}$, from which $\alpha$ is estimated to be $0.50 \pm 0.02$. The combined measurements provide four different ways to determine the two relevant parameters, indicating the consistency of the model.

to converge only for realization times $t_m > 10^5$. The increase of $\langle S(\omega, t_m) \rangle$ with time is directly related to the persistent property of the fBM[36].

We now focus on two important limits of our results, namely, the limits $2H \rightarrow 1$ and $\alpha \rightarrow 1$. In the first one, the process reverts to the traditional CTRW, for which the result is given by Eq. (10). Here, the two leading terms in the exact result for the PSD (Supplementary Eq. 33) converge to the same exponent yielding the simple asymptotic approximation $\langle S_{2H=1}(\omega, t_m) \rangle \approx 2(D+c)/\omega^2$. The agreement with Eq. (10) serves as a basic test to evaluate the results. The second limit ($\alpha \rightarrow 1$) is expected to converge to the known results for the standard fBM. In this limit, the PSD becomes (i) $\langle S(\omega, t_m) \rangle \sim 1/\omega^{1+2H}$ when $2H < 1$ and (ii) $\langle S(\omega, t_m) \rangle \sim t_m^{2H-1}/\omega^2$ when $2H > 1$. These expressions are in agreement with the known formulas for subdiffusive and superdiffusive fBM, respectively (see e.g.,[36]).

**Experimental results**. The derivation of the PSD of subordinated random walks enables us to characterize the motion of membrane proteins that typically interact with heterogeneous partners. These trajectories are obtained using single molecule tracking of labeled proteins in living cells. An example of a transmembrane protein that exhibits heterogeneous interactions is the voltage gated sodium channel Nav1.6. It was previously found that in the somatic plasma membrane of hippocampal neurons, Nav1.6 channels are transiently confined into cell surface nanodomains[63]. Because these nanodomains are only of the order of 100 nm in size, we can neglect the motion within an individual domain without altering the long time statistics of the process. Further, it was reported that the motion of these channels displays ergodicity breaking due to their transient confinement[64]. These effects lead to the idea of trapping and the CTRW type of dynamics. Thus, we model the confinement (immobilization) times using Eq. (5). An important property of heavy-tailed renewal processes is that they depend on the time that lapsed since the system started[65]. In the case of single molecule tracking of Nav channels, measurements start when the channel is delivered to the plasma membrane and, thus, the time $t = 0$ is well-defined. Besides transient immobilizations, Nav1.6 also show antipersistent fBM-like motion, leading to a non-linear time-averaged MSD. Here, we evaluate 87 Nav1.6 trajectories of 256 data points each, with a sampling time $\Delta t = 50$ ms.

Before digging into the PSD analysis of Nav channels, we consider their mean square displacement, which is a familiar statistical tool that helps us understand some basic properties of their motion. Furthermore, we can evaluate the validity of our

model for the motion of membrane proteins by analyzing the relations between the exponents that characterize the mean squared displacement and the power spectrum. Figure 4a shows the ensemble-averaged MSD (EA-MSD, $\langle x^2(t) \rangle$) together with its 95% confidence interval and the ensemble-average of the time-averaged MSD (EA-TA-MSD) for three different observation times, $t_m = 64\Delta t$, $128\Delta t$, and $256\Delta t$. The EA-TA-MSD is defined in its usual way,

$$\overline{\langle \delta^2(\tau, t_m) \rangle} = \frac{1}{t_m - \tau} \left\langle \int_0^{t_m - \tau} [x(t+\tau) - x(t)]^2 dt \right\rangle , \quad (16)$$

where, using the same notation as in the autocorrelation function, $\tau$ denotes the lag time. The difference between the EA-TA-MSD and the EA-MSD (Fig. 4a) is a direct indication of ergodicity breaking in the motion of Nav channels[30,64]. In the context of our model, the ergodic hypothesis breaks down since $\alpha < 1$. In theory, it should be possible to use the ensemble-averaged MSD to extract information about the exponents that characterize the motion. However, when the number of trajectories is not very large (as is usually the case in live cell experiments), the estimation of exponents from this metric is very poor due to statistical errors. This effect can be directly seen in the confidence interval of the MSD in Fig. 4a. Thus, we propose here to employ in addition to the TA-MSD a robust metric such as the PSD.

The EA-TA-MSD of the subordinated process scales as[66]

$$\overline{\langle \delta^2(\tau, t_m) \rangle} \sim \frac{\tau^{1-\alpha+2\alpha H}}{t_m^{1-\alpha}} . \quad (17)$$

We have measured both the EA-TA-MSD (Fig. 4a) and the PSD (Fig. 4b), with different observation times $t_m$. From the MSD, using Eq. (17), we extract exponents $\alpha = 0.54 \pm 0.02$ and $H = 0.32 \pm 0.08$. Remarkably, this is nearly identical to the estimation based on the PSD, where, using Eq. (13), we obtain $\alpha = 0.50 \pm 0.02$ and $H = 0.25 \pm 0.11$. The agreement is not a coincidence and it indicates that the underlying model of a subordinated process is consistent with two independent measurements. In other words, we can use one set of measurements (e.g., PSD) to predict the exponents of the other (e.g., MSD) and show that the selected model works. From a single set of data we cannot make this conclusion. Namely, if we record $\beta$ and $z$, we can easily estimate the exponents $\alpha$ and $H$, but that, as a stand alone, is not informative, since the number of fitting parameters (two) is the same as the number of linear equations given in the relations between the exponents ($\beta$, $z$) and the exponents ($\alpha$, $H$). Hence, extraction of these exponents with

an additional measurement is required to find a consistent theory, beyond merely fitting parameters.

A key aspect of these measurements is that the PSD is obtained for different observation times. By increasing the measurement time, we indeed observe the aging power spectrum, an effect that could have been missed, as the natural tendency in experiments is simply to use the longest available trajectories. The PSD decays with observation time, i.e., $z < 0$, as predicted for a process with Hurst exponent $H < 1/2$ (see Eq. 13). The PSD amplitude as a function of measurement time $t_m$ is shown in the inset of Fig. 4b, indicating $z = -0.50 \pm 0.02$. This spectral analysis in combination with the MSD confirms the predictions stating that the motion of Nav channels is a subordinated process and lets us obtain accurate estimates of the waiting time distribution and the Hurst exponent. While the goal of this work pertained to the dynamics of proteins, it is directly applicable to any process where a correlated random walk coexists with a non-ergodic CTRW.

## Discussion

We characterize subordinated random walks via two exponents, the Hurst exponent $H$ and the exponent that describes the heavy-tailed waiting time distribution $\alpha$. The Hurst exponent governs the correlations between increments and the memory effects of the random walk, while the exponent $\alpha$ is responsible for the long waiting times. We observe that the PSD is found to be accurately described by the formula $S(\omega, t_m) \sim \omega^{-\beta} t_m^z$, where the exponents $\beta$ and $z$ are uniquely defined by $H$ and $\alpha$ (see Eqs. 13 and 14).

Our results can be divided into two large classes depending on whether the increments are positively or negatively correlated, that is $H > 1/2$ or $H < 1/2$. The case $H < 1/2$ is associated with the tracer's interactions with a viscoelastic medium which lead to subdiffusion, while $H > 1/2$ is associated with persistent walks that can lead to superdiffusion, which is, in turn, related to active transport. Let us discuss first our results for antipersistent random walks ($H < 1/2$) because this is the relevant regime for the dynamics of the membrane proteins we studied. In this situation, $\beta = 2 - \alpha + 2\alpha H$, i.e, it is influenced by both the properties of the fBM and the CTRW, and it falls in the range $1 < \beta < 2$. In contrast, the exponent $z$ is dictated solely by the power law trapping times of the CTRW and it shows aging, that is $z < 0$. Specifically, the PSD decays with measurement time with an exponent $z = \alpha - 1$. As such, the aging process in this regime does not contain any information about the fBM. For the Nav1.6 ion channels we recorded aging power spectra with $z = -0.50$ and $\beta = 1.75$, from which we estimated the Hurst index $H$ of the fBM-like process and the exponent $\alpha$ that characterizes the tail of the distribution of immobilization times. Then, it is possible to use the measured exponents $z$ and $\beta$ (which give $\alpha$ and $H$) to predict the exponents of the time- and ensemble-averaged MSD, and compare these predictions to the experimental data. An agreement between predicted and measured exponents would show that the model is working well without any fitting to the MSD measurements. Indeed, the experimental results with Nav channels provide a very strong validation of our hypothesis, in which these channels can be described by an antipersistent random walk ($H < 1/2$) in the presence of traps caused by interactions with heterogeneous partners at the plasma membrane ($\alpha < 1$). The antipersistent walk is a signature of spatial heterogeneity and self similar obstructions in the membrane, while the heavy tailed waiting times are caused by trapping events, e.g. energy disorder. Our findings that subordinated fBM is the relevant model for ion channels is significant. The slow dynamics can rationalize the organization of these membrane proteins, as practically immobile, but still have some dynamics which is important for allowing interactions with cytoskeletal components and other reaction partners. We expect

that our model can be used to determine diffusion-limited reaction rates.

The case of positively correlated increments $H > 1/2$ leads to much richer phenomena, and we encounter both aging (a decay of the fluctuations) and rejuvenation (increase of the fluctuations) with measurement time. In this situation, $\beta$ is a constant $\beta = 2$ independent of the exponents of either the fBM or the CTRW. Note that this is the same frequency scaling as that of Brownian motion and the traditional CTRW. Nevertheless, the exponent $z$, given by Eq. (14), presents intriguing properties and, hence, it is the aging that informs about interesting physical effects. The smaller $\alpha$, the faster the fluctuations are inhibited over time. This effect is due to the particles becoming more and more immobile, i.e., they find deeper and deeper traps the longer the time that lapses since the preparation of the setup. However, when $H$ is increased, the fBM becomes more superdiffusive and, strikingly, the PSD can be observed to rejuvenate, i.e., the fluctuations become more prominent over time. Precisely, the turnover from aging to rejuvenating takes place when $H > 1/(2\alpha)$. Thus, one can infer the region in phase space to which the process belongs by noting whether the fluctuations increase or decay (see Fig. 3d for a full phase diagram). The special case $z = 0$, that is so often tacitly implied in the $1/f$ literature, is actually rare in subordinated diffusive processes and it only takes place when $H = 1/(2\alpha)$. Note, however, that normal Brownian motion takes place in the limit that $H = 1/2$ and $\alpha = 1$, which also implies $z = 0$, namely the absence of an aging effect.

Our analytical results describe the spectral content of a wide class of non-stationary processes with scale invariant correlation functions. The derivations are obtained using the aging Wiener–Khinchin theorem and we demonstrate the applicability of this theory with experimental trajectories of molecules in live cells. The class of processes that we study involves the coexistence of a fractional process with correlated increments and power-law distributed sojourn immobilization times. Beyond the motion of proteins, which was studied here in detail, these processes are encountered in vastly diverse scientific fields, such as hydrology[45,67] and movement ecology[68], and thus our results are expected to be widely applicable. The PSD analysis is very robust, particularly in noisy systems where it is impossible to obtain a very large number of experimental trajectories. Thus, the analysis is useful in elucidating the statistical properties of trajectories obtained by single-particle tracking in living cells, opening a new avenue in the analysis of protein transport.

## Methods

**Numerical simulations**. We performed all simulations in MATLAB. To generate a CTRW ($H = 1/2$), we synthesized increments drawn from a standard normal random variable, i.e., $\Delta x^2 = 1/2$. Subsequently, the times between steps were drawn from a Pareto distribution $\psi(t) = \alpha t^{-(1+\alpha)}$ for $t \geq 1$. For the subordinated random walk with $H \neq 1/2$, we obtained the increments using the MATLAB function wfbm to generate fBM. In this case, $\Delta x$ is a constant that depends on $H$. For each case, a total number of 10,000 realizations were obtained with either $t_m = 2^{16}$ or $t_m = 2^{18}$ and a sampling time of 1.

**Live cell imaging and single-molecule tracking**. Experimental details for cell culture, transfection, labeling, and imaging have been published previously[63]. Briefly, E18 rat hippocampal neurons were plated on glass-bottom dishes that were coated with poly-L-lysine. Neurons were grown in Neurobasal medium (Gibco/ Thermo Fisher Scientific, Waltham, MA, USA) with penicillin/streptomycin antibiotics (Cellgro/Mediatech, Inc., Manassas, VA, USA), GlutaMAX (Gibco/Thermo Fisher Scientific, Waltham, MA, USA), and NeuroCult SM1 neuronal supplement (STEMCELL Technologies, Vancouver, BC, Canada). For imaging, the cultures were incubated in imaging saline consisting of 126 mM NaCl, 4.7 mM KCl, 2.5 mM CaCl$_2$, 0.6 mM MgSO$_4$, 0.15 mM NaH$_2$PO$_4$, 0.1 mM ascorbic acid, 8 mM glucose, and 20 mM HEPES (pH 7.4). Neurons were transfected with a Nav1.6 construct containing an extracellular biotin acceptor domain (Nav1.6-BAD,[63]), using Lipofectamine 2000 (Invitrogen, Life Technologies, Grand Island, NY, USA). pSec-BirA (bacterial biotin ligase) was co-transfected to biotinylate the channel. Labeling of surface channels was performed before imaging. Neurons were rinsed with imaging

saline and then incubated for 10 min at 37 °C with streptravidin-conjugated CF640R (Biotium, Hayward, CA, USA) diluted 1:1000 in imaging saline. Total internal reflection fluorescence images were acquired at 20 frames/s using the 647 nm laser line of a Nikon Eclipse Ti fluorescence microscope equipped with a Perfect-Focus system, an Andor iXon EMCCD DU-897 camera, and a Plan Apo TIRF 100×, NA 1.49 objective. Imaging was performed at 37 °C using a heated stage and objective heater. Nav trajectories were obtained by single-molecule tracking using the U-track algorithm[69].

**Reporting summary**. Further information on research design is available in the Nature Research Reporting Summary linked to this article.

## Data availability

The datasets generated during the current study have been deposited in the Zenodo.org database under https://doi.org/10.5281/zenodo.5528301.

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

## Acknowledgements
The Nav1.6 imaging was performed by Dr. Elizabeth Akin. D.K. thanks Dr. Mike Tamkun for his help with the experiments and useful discussions. We acknowledge the support from the Colorado State University Libraries Open Access Research and Scholarship Fund (D.K.), National Science Foundation grant 2102832 (D.K.), and Israel Science Foundation grant 1898/17 (E.B.).

## Author contributions
D.K. conceived and designed the project, and constructed the theory. D.K and Z.R.F. generated numerical simulations and analyzed the data. D.K, E.B., and Z.R.F. interpreted the results. D.K. wrote the manuscript, assisted by E.B. and Z.R.F.

## Competing interests
The authors declare no competing interests.
