## [Peer Review File · Nature Communications]

Reviewers' Comments:

Reviewer #1:

Remarks to the Author:

The manuscript deals with the spectral analysis of non-stationary diffusive processes using the aging Wiener-Khinchin theorem. This class of processes involves the coexistence of correlated fractional Brownian motion and power-law distributed waiting times. The authors show that the power spectral density (PSD) exhibits a power-law scaling with an exponent that depends on the characteristics of both underlying processes. This analysis is then used by the authors to characterize the motion of voltage-gated sodium channels on the surface of hippocampal neurons.

The manuscript is well written and the analytical derivations seem to be sound. Its main result is the derivation of the PSD of subordinated processes by using hypergeometric functions, which can be approximated by simple power laws in the experimentally relevant frequency range. The authors show that the exponents H and α that determine subordinated processes can be obtained from the MSD and PSD. Different regimes of $H < 1/2$ (subdiffusive) and $H > 1/2$ (superdiffusive) are then investigated, and it is also shown that the PSD exhibits aging, namely that it depends on the experimental time t_m . These results are interesting and important to the field.

After having derived the PSD, the authors use their approach to characterize the motion of the voltage gated sodium channels in the somatic plasma membrane of hippocampal neurons. The experimental details have been previously published [60], where it has been shown that the motion is subdiffusive, and the exponent α was computed. The analysis done in the current manuscript allows one to find both the exponents H and α by simultaneously looking at the MSD and PSD of the data.

Major comments:

The generalization of the Wiener-Khinchin theorem to non-stationary processes has already been done in [2-4]. Therefore, although constituting a nice step forward, the derivation of the PSD cannot by itself warrant publication in Nature Communications. Furthermore, it is worth noting that one can find the exponents H and α of the experimental data by alternatively computing, in addition to the MSD, the ensemble-averaged time-averaged square displacement (EA-TA-SD), whose power-law scaling for subordinate processes is known. Thus, computing H and α for this data set does not require the theoretical formalism developed here by the authors for the PSD. As a result, since the main contribution of the manuscript - the analytical derivation of the PSD for subordinated processes - is rather technical and is beyond the reach of the general readership of Nature communications, I cannot recommend publication of the current manuscript in this journal. Nonetheless, I think it deserves publication in a more physics-oriented journal such as PRL or PRX.

Minor comments:

1. In some places the authors write "ensemble-averaged MSD". To be more accurate, they should instead write "ensemble-averaged square displacement" or "EA-SD", since there is no double averaging in this case. In addition, the authors write "EA-TA-MSD", which should in fact be "EA-TA-SD".
2. It is my understanding that the analytical results obtained by the authors are mostly relevant for large measurement times and/or frequencies such that $\omega * t_m \gg 1$. In this case, it is a bit superfluous to write the solution to the integrals in equations 13 and 24 using hypergeometric functions. Therefore, in my opinion, it would be better to directly write the asymptotic solutions (equations 14 and 25) in the main text, and keep the full expressions only in the supplementary information.

Reviewer #2:

Remarks to the Author:

The work "Aging power spectrum of membrane protein transport and other subordinated random walks" by Z.R. Fox, E. Barkai and D. Krapf is a very interesting and potentially important contribution. It presents a nice piece of theoretical work, and combines it with analysis of

experimental data showing an apparently similar behavior.

The theoretical part of the work investigates a specific model of nonstationary anomalous diffusion, namely a process subordinated to a correlated discrete random walk akin to fractional Brownian motion. For this process, analytical expressions are derived for aging spectral densities, and different regimes are investigated. This work represents a new, important step of work of one of the authors, Eli Barkai, on aging effects in spectra. In the experimental part of the work this behavior is compared with the data on motion of Nav1.6 channels, which was under continuous investigation in Diego Krapf's group, also participating present work. The corresponding references in the present work are Refs. [60-62].

The Ref. [62] is cited in connection with weak ergodicity breaking property of the corresponding process. This work however comes up with an apparently different model of the stochastic process governing the behavior of exactly the same system, namely with the two-state model in which the molecules alternate between free diffusion and confined motion. The model discussed in the theoretical part of the present work however does not exhibit two states (the only thing in common between the two models is the existence of prolonged intervals with relatively low mobility). Now the following questions arise, and have to be answered: Is there a possibility to distinguish in favor of one of the models? Has the earlier one to be abandoned? Or maybe the behavior is pertinent to a larger class of models, to which both, the new and the old one do belong? If yes, how broad is the class? Does a new model serve as a typical example or as a relatively detailed description? Can different models be distinguished on the base of spectral analysis, or other means are necessary? Knowing all this is important to assess the true usefulness of the approach in elucidating the statistical properties of experimental trajectories in live mammalian cells which the authors stress in the conclusions to the present work.

Reviewer #3:

Remarks to the Author:

This is an interesting manuscript about signal processing using statistical mechanics methods. The major claim is that the aging Wiener-Khinchin theorem can be used to derive the power spectral density of fractional Brownian motion coexisting with a scale-free continuous time random walk. There is a strong mathematics behind the paper and make it solid. The subject is novel and the conclusions are original. The authors also applied their methods into an experimental data previously published by some of the authors [ref 60]. However, the way it is written will not be on the interest of a broad scientific community. So, I suggest a more specific journal.

Please find below some comments the authors should consider.

Major problems:

The introduction takes too long to reach the focuses of the manuscript. It can be fixed by moving some of the content of the lines 62 to 70 to the beginning of the introduction.

The inline equation in the line 22 is only valid in the asymptotic regimes;

The sentence starting at the line 40 is a bit confusing. Which class of the protein they refer? where they observed the immobilization time? some references could be cited here;

At line 43, it is not clear the mean of "such diffusive transport";

The sentence that starts at line 46 apparently says the opposite of the first sentence of the same paragraph;

Fig 1: axis units are missing.

Several things are well detailed in the manuscript, however, the Hurst exponent are not.

Eq 8 and the following paragraph are tough to understand. Specifically the terms "first expectation" and "second expectation";

Fig 2: It is important here to clearly indicate what were the values of the parameters used in the simulation to follow the scientific principle that the results must be reproducible. For example, is the time in seconds or in units of t_0 ? It is also interesting that the authors give a notion of this time in physical units to determine whether the scale of time and frequency that they are adopting makes sense from an experimental point of view.

Eq 24: Function 2F3 not defined;

Minor points

Eq. 1, since "m" (in " t_m ") is an abbreviation of "measurements" it should not be in italic.

Line 58: "On" should be lower case;

Line 68/70: the authors mixed the words "analysis" and "model", I believe it should be one or the other.

Eq. 3 and Eq 16: at the end of the equation, replace "." by ","

Line 139: " C_{TA} " the "TA" should not be in italic.

In Figure 3b, the agreement is not as good as in the Figure a. The authors could explain why this is so.

Line 208: It is important to make it clear that it is not about transmembrane transport through the channels, but rather the movement of the channel itself in the membrane. Perhaps it is good at this point to comment that, in addition to the protein movement being described by a random walk, recent works also found that the movement of particles through biological channels can also be described by a random walk.

Line 213: "We have... [60]", but not all the authors of the present manuscript are authors of the ref 60. The "We" mislead the reader.

Reviewer #1:

R1.1 *The manuscript deals with the spectral analysis of non-stationary diffusive processes using the aging Wiener-Khinchin theorem. This class of processes involves the coexistence of correlated fractional Brownian motion and power-law distributed waiting times. The authors show that the power spectral density (PSD) exhibits a power-law scaling with an exponent that depends on the characteristics of both underlying processes. This analysis is then used by the authors to characterize the motion of voltage-gated sodium channels on the surface of hippocampal neurons.*

The manuscript is well written and the analytical derivations seem to be sound. Its main result is the derivation of the PSD of subordinated processes by using hypergeometric functions, which can be approximated by simple power laws in the experimentally relevant frequency range. The authors show that the exponents H and α that determine subordinated processes can be obtained from the MSD and PSD. Different regimes of $H < 1/2$ (subdiffusive) and $H > 1/2$ (superdiffusive) are then investigated, and it is also shown that the PSD exhibits aging, namely that it depends on the experimental time t_m . These results are interesting and important to the field.

After having derived the PSD, the authors use their approach to characterize the motion of the voltage gated sodium channels in the somatic plasma membrane of hippocampal neurons. The experimental details have been previously published [60], where it has been shown that the motion is subdiffusive, and the exponent α was computed. The analysis done in the current manuscript allows one to find both the exponents H and α by simultaneously looking at the MSD and PSD of the data.

Reply to R1.1 We thank the reviewer for carefully reading the manuscript and for highlighting the importance of this work.

Major comments:

R1.2 *The generalization of the Wiener-Khinchin theorem to non-stationary processes has already been done in [2-4]. Therefore, although constituting a nice step forward, the derivation of the PSD cannot by itself warrant publication in Nature Communications. Furthermore, it is worth noting that one can find the exponents H and α of the experimental data by alternatively computing, in addition to the MSD, the ensemble-averaged time-averaged square displacement (EA-TA-SD), whose power-law scaling for subordinate processes is known. Thus, computing H and α for this data set does not require the theoretical formalism developed here by the authors for the PSD. As a result, since the main contribution of the manuscript - the analytical derivation of the PSD for subordinated processes - is rather technical and is beyond the reach of the general readership of Nature communications, I cannot recommend publication*

of the current manuscript in this journal. Nonetheless, I think it deserves publication in a more physics-oriented journal such as PRL or PRX.

Reply to R1.2 We thank the reviewer for stating that our work deserves publication in such high impact journals as PRL or PRX. We have extensively modified the manuscript in order to address the concerns from the reviewer that led him/her to conclude that the article is better suited in a more physics-oriented journal. As explained in the text, and as noted by the referee, the aging Wiener-Khinchin theorem was already the subject of publications in the physics literature. So why publish this paper in Nature Communication? The previously published papers contain a very general relation, namely a connection between the non-stationary correlation functions and the aging PSD. However, this relation by itself does not yield any specific information nor does it show that it is applicable in real life experiments. Here, we meet this challenge, we compute the aging correlation function of a model which is widely applicable, and then show how it is related to experiments. The insight is that we can connect the exponents describing the random walk (H and α) with the exponents describing the PSD. We opted for Nature Communications not only due to its prestige, but rather because the journal caters to a wider audience. PSD of both stationary (standard Wiener-Khinchin) and scale invariant processes (aging Wiener-Khinchin) are widely applicable, though in the latter case we provide experimental proof here for the first time.

The reviewer advances an excellent point; the exponents H and α can be analytically obtained from both the ensemble-averaged MSD and the ensemble-time-averaged MSD. Here we give two answers to this query. First, in real experiments obtaining the exponents from the ensemble-averaged MSD is generally impossible due to the poor statistics associated with this metric when it is not feasible to obtain a very large number of trajectories. To put it differently, in most single molecule experiments, the evaluation of the ensemble-averaged MSD is rather poor, unless one has a very large ensemble (the same is not true with respect to the PSD). To prove this point, we direct the reader to figure 4a, which clearly shows the poor quality of the ensemble average. Thus, a second statistic such as the PSD is highly desirable. Second, from measurements of ensemble- and time-averaged MSD we gain information on two exponents. However, the input of the theory is also two exponents (α and H), and this as a stand-alone is hence merely a fitting procedure. The same might be claimed with respect to the PSD. Hence in the new version of the manuscript, we present information both from the MSD and the PSD; using the exponents from one measurement, one may predict without any fitting the outcome of the second measurement. This is a strong indication that the basic model is correct. These aspects are now explained in detail in lines 288-308

“In theory, it should be possible to use the ensemble-averaged MSD to extract information about the exponents that characterize the motion. However, when the number of trajectories is not very large (as is usually the case in live cell experiments),

the estimation of exponents from this metric is very poor due to statistical errors. This effect can be directly seen in the confidence interval of the MSD in Fig. 4a. Thus, we propose here to employ in addition to the TA-MSD a robust metric such as the PSD. ...”

“... The agreement is not a coincidence and it indicates that the underlying model of a subordinated process is consistent with two independent measurements. In other words, we can use one set of measurements (e.g., PSD) to predict the exponents of the other (e.g., MSD) and show that the selected model works. From a single set of data we cannot make this conclusion. Namely, if we record beta and z, we can easily estimate the exponents alpha and H, but that, as a stand alone, is not informative, since the number of fitting parameters (two) is the same as the number of linear equation given in the relations between the exponents (beta, z) and the exponents (alpha, H). Hence, extraction of these exponents with an additional measurement is required to find a consistent theory, beyond merely fitting parameters.”

and in lines 341-344

“...it is possible to use the measured exponents z and beta (which give alpha and H) to predict the exponents of the time- and ensemble-averaged MSD, and compare these predictions to the experimental data. An agreement between predicted and measured exponents would show that the model is working well without any fitting to the MSD measurements.”

We are sorry that the previous version failed to highlight some of the important aspects of this work beyond the calculation of the PSD and the technical aspects. In the revised manuscript, we send many technical aspects to the Supplementary Information, and we substantially reduced the number of equations in the main text, to make it within the reach of the general readership of Nature Communications. Further, we explain in detail the key findings of our work beyond the calculation of PSD, both from a theoretical point of view as well as the breakthrough in the analysis of a broad class of experimental data. These explanations concern the aging of the power spectrum, the relation between non-stationary processes and $1/f$ noise, the combination of the two most broadly used models for anomalous diffusion, and the use of the aging Wiener-Khinchin theorem in experiments. In a nutshell, similar to the Wiener-Khinchin theorem, which gives the PSD in terms of a correlation function, but says nearly nothing about the correlation function itself, our work explains (for the new aging Wiener-Khinchin theorem) what the correlation function is for an important process, thus relating the exponents of the PSD and the characteristics of the diffusion process, as mentioned. We now highlight in the introduction (lines 34-37 and 42-53)

“Notwithstanding previous advances, many questions remain open. First, the aging Wiener-Khinchin theorem relates the aging power spectrum with $z \neq 0$ ” (z is the aging

exponent PSD $\sim \omega^{(-\beta)} t_m^z$) “to a non-stationary correlation function (soon to be discussed). However, how can one find this correlation function?”

“...both models, when standing alone, are usually non-sufficient to describe the transport of particles that alternate between a trapping phase (like in CTRW) and correlated motion (like in fBM), as is the case in live cells, for example due to interactions in a viscoelastic medium [26]. The open questions begin with how to create a marriage between these models? Then, can we obtain the correlation functions and 1/f spectrum? ... these goals can elucidate whether the whole approach to the PSD is useful in experiments. Specifically, we demonstrate the applicability of aging Wiener-Khinchin theorem and the corresponding calculation of the correlation function with experimental recording of the power spectra of the motion of ion channels in the plasma membrane of mammalian cells.”

and in lines 71-77

“Following previous work, we promote a theory that shows how the most basic formula of 1/f noise needs modifications, namely that $S(\omega) \sim \omega^{(-\beta)} t_m^z$ as mentioned. The question that still needs to be addressed is what the physical meaning of the new exponent z is, to explore cases where it is negative (corresponding to a decrease of the PSD with time and, hence, aging) and cases where it is positive (corresponding to a PSD increasing with time and, hence, rejuvenation). Further, beyond the development of the theory, it is important to show how these effects are found experimentally.”

The proposed questions are then answered throughout the text both in the results and in the discussion sections.

Finally, large sections of the manuscript have been rewritten so that it can be enjoyed by a large audience of Nature Communication readers interested in diffusion processes.

Minor comments:

R1.3 In some places the authors write "ensemble-averaged MSD". To be more accurate, they should instead write "ensemble-averaged square displacement" or "EA-SD", since there is no double averaging in this case. In addition, the authors write "EA-TA-MSD", which should in fact be "EA-TA-SD".

Reply to R1.3 The reviewer is correct that the use of ensemble-averaged square displacement should be sufficient to convey the meaning. We have given the request from the referee a great deal of thought and we have decided to keep the MSD notation instead of SD for the following reasons. (i) The use of ensemble-averaged mean square displacement (EA-MSD) is the standard nomenclature used in the field and deviating from this notation would be confusing to many

readers. (ii) While further discussion on the topic is possible, we assess that in this case the phrase ensemble-averaged is used as an adjective for how the average of the MSD is made. Therefore, it is grammatically correct. (iii) The word 'mean' is indeed redundant, but the acronym MSD has gained so much popularity that a very large number of readers understand its meaning without the need to read the definition in its first appearance. For example, lectures on diffusion processes typically refer to the mean squared displacement simply by its acronym MSD. This familiarity would not be true for a defined acronym SD and would make the manuscript more difficult to read. The same reasons hold for the EA-TA-MSD.

R1.4 *It is my understanding that the analytical results obtained by the authors are mostly relevant for large measurement times and/or frequencies such that $\omega \cdot t_m \gg 1$. In this case, it is a bit superfluous to write the solution to the integrals in equations 13 and 24 using hypergeometric functions. Therefore, in my opinion, it would be better to directly write the asymptotic solutions (equations 14 and 25) in the main text, and keep the full expressions only in the supplementary information.*

Reply to R1.4 The suggestion of the reviewer is excellent, and we have sent all the exact equations with hypergeometric functions to the Supplementary Information. We believe this change makes the manuscript much easier to read, while keeping the details in the SM for the specific readers interested in them.

Reviewer #2:

R2.1 *The work "Aging power spectrum of membrane protein transport and other subordinated random walks" by Z.R. Fox, E. Barkai and D. Krapf is a very interesting and potentially important contribution. It presents a nice piece of theoretical work, and combines it with analysis of experimental data showing an apparently similar behavior.*

The theoretical part of the work investigates a specific model of nonstationary anomalous diffusion, namely a process subordinated to a correlated discrete random walk akin to fractional Brownian motion. For this process, analytical expressions are derived for aging spectral densities, and different regimes are investigated. This work represents a new, important step of work of one of the authors, Eli Barkai, on aging effects in spectra. In the experimental part of the work this behavior is compared with the data on motion of Nav1.6 channels, which was under continuous investigation in Diego Krapf's group, also participating present work. The corresponding references in the present work are Refs. [60-62].

Reply to R2.1 We thank the reviewer for expressing interest in our work and appreciating the importance of the manuscript.

R2.2 *The Ref. [62] is cited in connection with weak ergodicity breaking property of the corresponding process. This work however comes up with an apparently different model of the stochastic process governing the behavior of exactly the same system, namely with the two-state model in which the molecules alternate between free diffusion and confined motion. The model discussed in the theoretical part of the present work however does not exhibit two states (the only thing in common between the two models is the existence of prolonged intervals with relatively low mobility). Now the following questions arise, and have to be answered: Is there a possibility to distinguish in favor of one of the models? Has the earlier one to be abandoned? Or maybe the behavior is pertinent to a larger class of models, to which both, the new and the old one do belong? If yes, how broad is the class? Does a new model serve as a typical example or as a relatively detailed description? Can different models be distinguished on the base of spectral analysis, or other means are necessary? Knowing all this is important to assess the true usefulness of the approach in elucidating the statistical properties of experimental trajectories in live mammalian cells which the authors stress in the conclusions to the present work.*

Reply to R2.1 The first question of the reviewer is due to our lack of explanations in the previous manuscript. We apologize for this poor judgement from our part and thank the reviewer for bringing this up. In the current work we approximate the free (anomalous) diffusion / confined motion two-state model as being free (anomalous) diffusion / immobile. Thus, we approximate the second state as being immobile because the confinement takes place in a very small nanodomain. This is now thoroughly explained in lines 265-269

“Because these nanodomains are only of the order of 100 nm in size, we can neglect the motion within an individual domain without altering the long time statistics of the process. Further, it was reported that the motion of these channels displays ergodicity breaking due to their transient confinement [64]. These effects lead to the idea of trapping and the CTRW type of dynamics.”

Regarding the question ‘Can different models be distinguished on the base of spectral analysis, or other means are necessary?’ In order to distinguish between different models, we would need to find the predictions for the other models that would be evaluated. What we find is that the model in the manuscript is consistent with the data and we explain this agreement in terms of using the combination of both statistics, the MSD and PSD. As mentioned in an answer to a query from Reviewer 1, in the new version of the manuscript, we present information both from the MSD and the PSD; using the exponents from one measurement, one may predict without any fitting the outcome of the second measurement. This is a strong indication that the basic model is correct. These aspects are now explained in detail in lines 299-308

“The agreement is not a coincidence and it indicates that the underlying model of a subordinated process is consistent with two independent measurements. In other words, we can use one set of measurements (e.g., PSD) to predict the exponents of the other (e.g., MSD) and show that the selected model works. From a single set of data we cannot make this conclusion. Namely, if we record beta and z, we can easily estimate the exponents alpha and H, but that, as a stand alone, is not informative, since the number of fitting parameters (two) is the same as the number of linear equation given in the relations between the exponents (beta, z) and the exponents (alpha, H). Hence, extraction of these exponents with an additional measurement is required to find a consistent theory, beyond merely fitting parameters.”

and in lines 341-344

“...it is possible to use the measured exponents z and beta (which give alpha and H) to predict the exponents of the time- and ensemble-averaged MSD, and compare these predictions to the experimental data. An agreement between predicted and measured exponents would show that the model is working well without any fitting to the MSD measurements.”

Reviewer #3:

R3.1 *This is an interesting manuscript about signal processing using statistical mechanics methods. The major claim is that the aging Wiener-Khinchin theorem can be used to derive the power spectral density of fractional Brownian motion coexisting with a scale-free continuous time random walk. There is a strong mathematics behind the paper and make it solid. The subject is novel and the conclusions are original. The authors also applied their methods into an experimental data previously published by some of the authors [ref 60]. However, the way it is written will not be on the interest of a broad scientific community. So, I suggest a more specific journal.*

Reply to R3.1 We thank the reviewer for noting the interesting aspects of the manuscript, its novelty, rigor, and originality.

We believe that the criticism made here is fair, and indeed similar to what was mentioned by Reviewer 1. Hence, as we stated above, we dramatically updated the manuscript. We have rewritten whole sections, and we hope the reviewer finds this revision suitable for the style of Nature Communications. In particular, we have rewritten the introduction section, most of the experimental results, and the discussion. In addition, we have modified the mathematical sections where we have sent many mathematical details to the Supplementary Information. The overall number of equations in the revised manuscript was greatly reduced.

Major problems:

R3.2 *The introduction takes too long to reach the focuses of the manuscript. It can be fixed by moving some of the content of the lines 62 to 70 to the beginning of the introduction.*

Reply to R3.2 We appreciate the reviewer brought this point to our attention. We have now rewritten the introduction and we brought the main focus of the manuscript to the first paragraph. Namely, in lines 30-37

“These developments, in turn, motivated a new theoretical framework, called aging Wiener-Khinchin theorem [20-22]. This new theorem replaces the celebrated Wiener-Khinchin theorem valid for stationary processes, which is widely applicable to systems that do not exhibit 1/f noise [23].

Notwithstanding previous advances, many questions remain open. First, the aging Wiener-Khinchin theorem relates the aging power spectrum with $z \neq 0$ ” (z is the aging exponent PSD $\sim \omega^{(-\beta)} t_m^z$) “to a non-stationary correlation function (soon to be discussed). However, how can one find this correlation function?”

R3.3 *The inline equation in the line 22 is only valid in the asymptotic regimes.*

Reply to R3.3 We corrected this inline equation now in line 91.

R3.4 *The sentence starting at the line 40 is a bit confusing. Which class of the protein they refer? where they observed the immobilization time? some references could be cited here.*

Reply to R3.4 We have deleted this sentence. Instead, we write in lines 57-59

“Recent molecular dynamics simulations in combination with previous experimental results have shown that the internal dynamics in globular proteins are self-similar and the autocorrelation function is aging over an astonishing 13 decades in time [2, 28].”

and in lines 61-62

“...this behavior is widespread and found from the dynamics of proteins within cell membranes to the scaling behavior of heartbeat time series [27, 29].”

R3.5 *At line 43, it is not clear the mean of “such diffusive transport”.*

Reply to R3.5 We replaced this sentence with the following statements in lines 38-45

“In the context of diffusion in cells as well as in many other complex systems, Mandelbrot’s fractional Brownian motion (fBM) [24] and the Montroll-Weiss continuous

time random walk (CTRW) [25] are two widely investigated models of anomalous transport. While the fluctuations in fBM are stationary, the CTRW process is inherently non-stationary. However, both models, when standing alone, are usually non-sufficient to describe the transport of particles that alternate between a trapping phase (like in CTRW) and correlated motion (like in fBM), as is the case in live cells, for example due to interactions in a viscoelastic medium [26]."

R3.6 *The sentence that starts at line 46 apparently says the opposite of the first sentence off the same paragraph.*

Reply to R3.6 We apologize for the lack of clarity in this paragraph. We have rewritten it and hopefully it is now clear.

R3.7 *Fig 1: axis units are missing.*

Reply to R3.7 We added units to the figure.

axes are unitless. For clarity we have added additional labels to the axes of Fig 1: "sampling time units" to x axis and "arbitrary units" to y axis. The figure presents numerical simulations where the sampling time is unity and the simulation waiting times are drawn from Pareto distribution where the minimum time is also unity. We also added a specific statement about this issue in the methods (Numerical Simulations, lines 390-392) and in the caption of Fig. 2

"The times and displacements are unitless, i.e., the simulation sampling time is 1."

R3.8 *Several things are well detailed in the manuscript, however, the Hurst exponent are not.*

Reply to R3.8 We have now explained the meaning of the Hurst exponent. The definition of the Hurst exponent appears in lines 163-164

"We consider a fBM-like process at discrete times, $n = 0, 1, 2, 3, \dots$, with Hurst exponent H , such that its autocorrelation function at the discrete times n is given by [24]

Equation (4)"

and an explanation of the meaning of the Hurst exponent is given in lines 322-324

"The Hurst exponent governs the correlations between increments and the memory effects of the random walk,..."

R3.9 Eq 8 and the following paragraph are tough to understand. Specifically the terms “first expectation” and “second expectation”

Reply to R3.9 We have reworded the explanation of that Equation (now Equation 6).

R3.10 Fig 2: It is important here to clearly indicate what were the values of the parameters used in the simulation to follow the scientific principle that the results must be reproducible. For example, is the time in seconds or in units of t_0 ? It is also interesting that the authors give a notion of this time in physical units to determine whether the scale of time and frequency that they are adopting makes sense from an experimental point of view.

Reply to R3.10 The specific values of all the numerical simulations are now provided in the Methods section (lines 388-395) and units were added to the time axis.

In addition to stating the details of the parameters of the numerical simulations, we implemented some changes to the manuscript, which we believe will help understand the different units of the constants involved. Specifically, we use $(\Delta x)^2$ for the scaling constant in Equation 4, which has units of m^2 . We also use D for the constant in Equation 14, which is a generalized diffusion coefficient with units of m^2/s^γ , where $\gamma=2H$. The details of how this constant is obtained are in the Supplementary Information, showing that the ensemble-averaged MSD is $\langle x^2 \rangle = 2 D t^\gamma$.

Regarding experimental data, the specific values of t_0 and Δx are not uniquely defined. As in normal diffusion, these values can be made as small as needed. Usually in Brownian motion, t_0 is the mean free time, but any combination of such that $(\Delta x)^2/t_0 = 0$ will give the same results. Similarly in the subordinated process, any combination that yields $(\Delta x)^2/t_0^\gamma = D$ will lead to the same results and as such, the specific value of t_0 is not critical. This fact is also true for the PSD, see e.g., (i) the PSD amplitude in Equation 10 is proportional to $(\Delta x)^2/t_0^\alpha$ when $H=1/2$, (ii) the PSD amplitude in Equation 14 is proportional to c , which is proportional to $(\Delta x)^2/t_0^\gamma$ when $H<1/2$, and (iii) the PSD amplitude in Equation 14 is proportional to D , which is also proportional to $(\Delta x)^2/t_0^\gamma$ when $H>1/2$.

R3.11 Eq 24: Function 2F3 not defined.

Reply to R3.11 Thanks for noticing this omission. We have now moved the equation dealing with this function to the Supplementary Information and, hence, the function is now defined right after Supplementary Equation 32.

Minor points

R3.12 Eq. 1, since “m” (in “t_m”) is an abbreviation of “measurements” it should not be in italic.

Reply to R3.12 We corrected t_m throughout the manuscript and now the m is Roman.

R3.13 Line 58: “On” should be lower case.

Reply to R3.13 The sentence was deleted in the revised manuscript.

R3.14 Line 68/70: the authors mixed the words “analysis” and “model”, I believe it should be one or the other.

Reply to R3.14 The sentence was deleted in the revised manuscript.

R3.15 Eq. 3 and Eq 16: at the end of the equation, replace “.” by “,”

Reply to R3.15 We paid careful attention to the endings of the equations in the revised manuscript.

R3.16 Line 139: “<C_{TA}>” the “TA” should not be in italic.

Reply to R3.16 We corrected the format of the TA abbreviation subscript.

R3.17 In Figure 3b, the agreement is not as good as in the Figure a. The authors could explain why this is so.

Reply to R3.17 The figure is now in the Supplementary Information. The comment of the reviewer made us evaluate the results for a larger range to make sure we did not have any problems there. The autocorrelation functions are now presented in the range for lag times between 1 and 10,000 instead of being from 10 to 500 as previously done. Most likely the deviations are due to the realization time not being long enough, which shows up when the process exhibits long-range dependence, i.e., $H > 1/2$. We mention this issue in the Supplementary Information as well.

R3.18 Line 208: It is important to make it clear that it is not about transmembrane transport through the channels, but rather the movement of the channel itself in the membrane. Perhaps it is good at this point to comment that, in addition to the protein movement being described by

a random walk, recent works also found that the movement of particles through biological channels can also be described by a random walk.

Reply to R3.18 We have explained this in the revised version in lines 259-261

“... the motion of membrane proteins that typically interact with heterogeneous partners. These trajectories are obtained using single molecule tracking of labeled proteins in living cells.”

R3.19 Line 213: *“We have... [60]”, but not all the authors of the present manuscript are authors of the ref 60. The “We” mislead the reader.*

Reply to R3.19 This was corrected (line 263)

Reviewers' Comments:

Reviewer #1:

Remarks to the Author:

The authors have satisfactorily addressed all my comments and concerns, and have significantly revised the manuscript. They have practically rewritten most of the Introduction and Discussion sections, as well as many parts in the Results section. These revisions, along with moving many technical aspects from the main text to the Supplementary Information file, make the current version much more suitable for the general readership of Nature Communications. As a result, I now revise my original recommendation, and recommend publication of the manuscript in Nature communications as is.

Reviewer #2:

Remarks to the Author:

The authors have strongly rewritten the work, and improved considerably its readability. They have satisfactorily responded to my criticisms and incorporated the corresponding explanations in the new version of the text. Since I found already the previous version of the manuscript to be scientifically sound and interesting, I can now wholeheartedly recommend its publication.

Reviewer #3:

None